# What’s Your Pension Story? Women’s Perspectives during the COVID-19 Pandemic on Their Old-Age Pension Status, Past and Present

**DOI:** 10.3390/ijerph20105912

**Published:** 2023-05-22

**Authors:** Anat Herbst-Debby

**Affiliations:** The Gender Studies Program, Bar-Ilan University, Ramat Gan 5290002, Israel; anat.herbst@biu.ac.il

**Keywords:** COVID-19, inequality, gender, old-age pension policy, divorce, economic retirement status, life course, economic abuse

## Abstract

This study examines the present and retrospective views of mothers who are nearing or are at retirement age regarding their economic status, pension planning, and perceptions of state pension policy. The paper addresses gaps in the literature on the cross-intersections of employment history, vulnerable economic retirement status, and marital and parental status, thereby adopting a life course perspective. Based on in-depth interviews of thirty-one mothers (ages 59–72) during the COVID-19 pandemic, the findings revealed five themes—economic abuse: an unequal distribution of pension funds following divorce; regrets over past choices; COVID-19 and pensions; the state’s responsibility for old-age economic security; and knowledge is important, and I can help others. The study concludes that the majority of women at these ages perceive their current economic situation as a product of insufficient familiarity with pension plans, while voicing opinions about the state’s irresponsibility regarding people of retirement age.

## 1. Introduction

The aim of this study was to hear the stories of mothers who are nearing or are at retirement age about their employment pension and economic status, and to examine their perceptions of this in the present, as well as in retrospect, against the background of the COVID-19 pandemic. This paper adopts a life course perspective [1,2,3] to address the gap in the literature on the cross-intersections of employment history, vulnerable economic retirement status, and marital and parental status. The life course perspective allows us to view life events as a whole, and to understand the impact of these events upon women’s lives, by looking at the outcomes of temporal shifts in life [4] and other major life experiences, such as changing participation in the labor market, divorce, and economic abuse. This perspective can reveal how these experiences intersect with people’s vulnerable economic status in retirement, particularly as life course outcomes are often stimulated by the accumulation of disadvantages, as early life hardships can be compounded over time [5]. Moreover, vulnerable economic status in retirement can be linked to health problems [6,7].

The majority of older women, especially older mothers, suffer from financial insecurity and disadvantage [8,9,10]. This situation is compounded by union dissolution, which is considered one of the risky life course events that leads to economic hardship. Studies have shown a significant decline in the economic wellbeing of women (and children) during the post-divorce period [11,12,13,14,15,16]. Moreover, marital disruption has been found to have extensive health implications and psychological outcomes, as well as affecting quality of life [17]. Economic abuse by the husband has also been associated with the economic deterioration of women [18,19,20]. This article looks at these life experiences together, analyzing how they intersect with the economic status of women at retirement age.

Israel is an important case study from which we can learn beyond the specific case. Israel has the highest birth rate among the OECD countries [21], coupled with a welfare policy that gives little social protection to children and their families [22]. Moreover, Israel has implemented significant pension reforms that have led to the transfer of fiscal responsibility and risk during retirement to the individual. Groups that are more vulnerable, including women and individuals of a low economic status, pay a heavier price for these reforms [23]. This raises two research questions: (1) What is the retrospective and present view of older mothers regarding their pension planning? (2) What are mothers’ perceptions of the state pension policy?

## 2. Literature Review

### 2.1. Genderedness of Old-Age Vulnerability

The financial insecurity that many women in different societies, especially mothers, face in retirement has been well documented [8,9,10,24,25]. Their risk of falling into poverty is higher than for men [26,27,28,29,30]. Lower income at retirement age leads to a higher degree of social isolation and worsens housing and health conditions [31]. Moreover, the longer life expectancy of women leads to an increased risk of suffering from chronic disease, which raises their own costs of healthcare in old age [32,33].

The gender gap in pensions is related to disparities between men and women in the labor market. While the lack of economic stability in retirement is often ascribed to women’s low financial literacy regarding money management in general and in pension funds in particular [34], this assumption disregards the price that women have to pay throughout their life course due to structural barriers. Thus, there is structural discrimination against women in terms of salary, promotions, and work conditions. Moreover, overall, women do more invisible and unpaid work, and work fewer paid hours [30,35,36,37]. They usually invest more time in the social role of a primary caretaker of their parents, children, and other relatives. Caring obligations often lead to career interruptions, work breaks, and job reductions, which can therefore have a negative impact on their retirement income [35,38,39]. Investing in childcare at a young age can also lead to lower pension entitlements [40,41,42].

Feminist and health economists have pointed to the financial penalties of being a caregiver [43,44], including withdrawal from the labor force [45] and its associated costs, such as loss of wages and pensions, which are particularly substantial for women [46]. Indeed, mothers pay the price of childcaring throughout their life course, long after their offspring have left early childhood [47]. In particular, when compounded with low pay, this makes it very difficult for women to accumulate sufficient savings for a safe economic retirement [38,48]. Thus, despite a significant rise in their labor market participation, women in general, and mothers in particular, accumulate less in terms of employment pensions, and depend more than men on benefits provided by the state. In other words, their increased presence in the working world has not provided significant benefits to their financial stability at retirement age [10].

At the same time, reforms in welfare-state institutions in many countries have led to pension marketization, i.e., the outsourcing of portions of the public pension to market-based schemes [49,50]. This privatization has transferred most of the responsibility and accountability for the financial management of old age—including decisions made about and the risk involved in pension savings—from the state and employer to the individual [23,51]. Pension reforms in most OECD countries have ignored women’s difficulties in amassing an adequate retirement income [52,53]. Studies suggest that these changes actually entrench gender and income inequalities [53,54]. Financial difficulties are particularly common among those without occupational pension benefits, which affects women more than men [55]. The problematics underlying this change was accentuated in Israel, as the management of pension funds was transferred from the non-profits to the private companies in the business sector [23].

### 2.2. Divorce, Widowhood, and Mothers’ Vulnerability

Divorce is one type of life course transition, i.e., a period of rapid change when individuals redefine one or more of their social roles and move from one phase to another [15]. It is considered a risky life event in terms of the economic wellbeing of women and children, often leading to short- and long-term negative financial consequences [11,12,13,14,15,16]. Women are nearly always economically weaker after union dissolution (as are men, but to a lesser extent), due to an inability to overcome the loss of the partner’s income [12,14,56,57,58,59]. 

There are two main explanations for the abovementioned economic implications of divorce. The first is the lower wages of women stemming from gender discrimination in the labor market [60,61,62]. The second is the mother’s main responsibility for childcare [63], as well as the sociocultural expectations and practices stemming from this responsibility [64]. This is especially true in Israel, where motherhood has been constructed as a national mission [65,66]. Transition to parenthood increases the risk of poverty more than the transition to marriage or cohabitation [67,68]. This is reflected in the motherhood wage penalty [69], which impacts their pension accumulation [40]; although men raising children alone also pay a penalty, it is much higher for single mothers than for single fathers [13,70]. 

For women, poverty is embedded not only in divorce [15], but also in widowhood [71] due to a lack of access to the wages of a male breadwinner. Widows have a higher risk of economic deprivation and financial strain [72] than both married women and widowers [27,73,74]. On average, older widows (age 50 and older) have lower wealth holdings, income, and food expenditures, as well as poorer health, than married women in the same age category [74]. 

### 2.3. Vulnerability in the Wake of the COVID-19 Pandemic

The COVID-19 pandemic has posed multifaceted health, economic, financial, and social consequences [75,76,77], widening long-standing inequities (see reviews in [77,78,79]). Lockdowns and closures have deepened gender inequality, as loss of employment has been higher in sectors with high female employment rates [80,81,82,83]. Moreover, during the pandemic, employment for older US workers (ages 62–70) dropped substantially more than for younger individuals [84]. Those who experienced job loss have had a greater prevalence of financial hardship, food insecurity, and poor/fair self-rated health [85]. Loss of employment has also led to a rise in retirement among older individuals and women [86]. Early retirement can lead to losses in future retirement savings, thereby reducing pension benefits, and raising income inequality in one’s later years [87]. A recent study in New Zealand [88] suggested that, in the wake of the pandemic, women will have more trouble restoring their depleted savings than men, will be more affected by lower employment opportunities, and will also be more dependent on their pension benefits. The authors expect older female poverty to worsen considerably in the post-COVID years. 

### 2.4. Economic Abuse: IPV (Intimate Partner Violence) against Women

One of the major impacts of COVID-19, in addition to its health and financial consequences, has been the toll of the prolonged lockdowns and job loss on family cohesion, which has been reflected in the increased rates of both divorce and domestic violence against women, including economic abuse [89,90]. IPV, as manifested in economic abuse, involves practices aimed at maintaining power and control by placing obstacles to the partner’s—usually the woman’s—economic independence. The male abuser maintains control of the family finances, deciding how money will be saved or spent, and forcing his partner to depend on him to meet her personal financial needs [91]. This concept embraces control of money, goods, and assets (such as placing her on a strict allowance [92,93]; and keeping her “unbanked” (lacking a checking or savings accounts) and isolated from household financial information [94]. These practices affect the woman’s ability to manage, obtain, use, and maintain her economic resources independently (without supervision), threatening her financial security, and her potential to support herself [18,19,20]. Another type of economic abuse is related to income allocation, which is usually carried out through power relations operating within the household, based on different gender expectations [95,96]. 

There is a strong correlation between economic abuse and divorce [97]: abused women are 1.7–5.7 times more likely to end their marriage [98]. In some cases, the abuse continues, or even worsens after separation [99]. For instance, despite the rule of monetary sharing between spouses in the civil courts in Israel, women who experience divorce still receive, on average, only about a third of the joint property, even though by law they are entitled to at least half [100].

### 2.5. The Israeli Case

Israel represents a unique site for analyzing the economic standing of women. It has a dominant conservative family model, with lower divorce rates and higher marital rates compared to the other OECD countries [101,102]. Israeli culture attaches a considerable importance to Jewish family life in general, and childbirth in particular [65,66,103,104]. It is a traditional society that embraces robust family values [105,106]. By the ages of 45–59, about 90 percent of Israelis have been married and are parents [101]. Indeed, Israel has the highest birth rate in the OECD, with an average of 3.1 children per woman, compared to the OECD average of 1.7 [21]. At the same time, the employment rate of mothers, especially mothers of young children, is also high: 75 percent, compared to an average of 71 percent in the OECD countries [107].

This country has traditionally held a pronatalist welfare policy. In the early years of the Israeli state, a special grant was given to women who had ten or more children [108]. Jewish mothers have played an inherent role in Israeli nation-building, and the welfare allowances accorded to them are meant to help achieve that goal. At the same time, Israel’s welfare policy has been marked more as a concern for national-demographic goals, and less with women as individuals who require social benefits [109].

Despite the pronatalist policy and the laws encouraging women to bear many children, the state does not provide sufficient support to raise them [22]. The poverty rate of children is the highest among the OECD countries: 22 percent versus an OECD average of 13 percent [110]. Moreover, Israel is characterized by large income gaps between population sectors; for instance, the rate of low earners (up to two-thirds of median earnings) among full-time workers is 22.4 percent, compared to an OECD average of 14.4 percent [111]. Inequality is also embedded in the differences between family structures. The likelihood of divorce has increased in the last two decades among couples with a lower socioeconomic status in Israel, as in many other countries [112,113,114]. 

The sources of retirement Income in Israel are three-fold (Table 1 and Table 2): a residency-based basic old-age pension from the National Insurance Institute; an employment pension based on the individual’s earnings; and an optional private pension based on the individual’s personal savings [50]. A number of reforms carried out over the past few decades in Israel have impacted retirement income, particularly among the vulnerable groups such as mothers.

With respect to the state’s basic old-age pension, reforms have raised the retirement age and the age of eligibility for the allowance. The enactment of the Retirement Age Law in 2004 and the revision of the National Insurance Law (consolidated version) in 1995 led to a gradual rise in the age of the woman’s eligibility for old-age pensions, reaching the age of 67. At the time of the study, the minimum age of entitlement with income tests was 62 for women and 67 for men, and age 70 for both genders without income tests, respectively. Moreover, an individual’s employer can terminate employment without this being considered as age discrimination at age 67 for both men and women, although employment can continue as long as both parties are interested. In addition, in an attempt to protect wage earners who were not insured under any pension arrangement (about one-third of the civilian labor force [115]), the enforcement of a collective agreement addendum in 2008 required both employers and employees to start paying pension contributions. This is in keeping with most OECD countries, which have instituted mandatory pension savings [116]. Thus, employment pensions have moved from a defined benefit (DB) plan to a defined contribution (DC) plan that imposes higher costs on savers, including investment risk, individual mortality risk, cohort mortality risk, salary risk, and job tenure risk [50,54]. Previously, Israeli public sector workers and working members of the big labor unions anchored in collective agreements, and enjoyed the broad benefits of civil service employment pensions which covered all their pension payments. All other employers were not required to make mandatory pension contributions.) Moreover, pension annuities are calculated according to the life expectancy, which subsequently decreases the monthly benefits for retired women [117]. This is further aggravated by early retirement, which provides an occupational pension allowance at a lower monthly rate due to the greater number of expected years of life.

Fund privatization has significantly increased management fees, with the weakest workers paying the highest fees [23] and has led to the investment of pension funds in the stock exchange, thereby reducing the basic yield markedly, which used to be guaranteed by the state via designated bonds [117]. At the same time, individual savings in market-based pensions, which have grown in importance in respect to the pension income of all earners, negatively impact the vulnerable lower-income earners, a group most pronounced among mothers [23,54,117]. Similarly, the working poor, who once could receive a pension supplemented by an income support allowance in old age, are now required to make employee pension contributions from their meager wages [118]. The addition of tax credits for pension savings provides little relief for low-income employees who do not reach the tax threshold (37 percent of working men and 61 percent of working women, respectively [118,119]. 

## 3. Method

To address these research questions, a qualitative approach was developed that focused on the voice and experience of older mothers and aimed at revealing their perceptions of their economic status at retirement age. The goal of conducting constructive qualitative research is to reach an understanding of the meaning of the phenomenon being studied. It does not presume to present any comprehensive scientific truth, objective or absolute, but rather to examine the question of meaning within a concrete context [120].

Specifically, the current study employed semi-structured in-depth interviews, as is common in the feminist literature, to raise and explore women’s ideas, experiences, feelings, and sensations in their own words, rather than those of the researcher [121]. It is part of a larger research project involving the gender-critical analysis of the status of women in general, and mothers in particular, from different classes, and of diverse family statuses during their retirement.

A sample of thirty-one mothers was recruited through Facebook and WhatsApp social networks, as well as via human resource managers in varied workplaces. In the search for interviewees, women with a significant employment history in the past or present who were concerned with the question of their future pension were requested for interview. Women from a variety of occupations, employment histories, and educational fields, along with a wide spectrum of incomes, were also incorporated in this search and interviewed. The search was subsequently stopped when the point of saturation was reached. Following Small [122] (p. 28), the approach for this study called for “logical rather than statistical inference, for case rather than sample-based logic, for saturation rather than representation.”. Thus, though the sample was neither representative nor statistically large enough from which to generalize [123], statistical representativeness was not relevant for the purposes of this study (see [122,124]).

In the final sample, only Jewish women close to or after formal retirement age were included (age range = 59–72), to ensure that they had already worked enough years to afford them close to the maximum number of benefits available. The decision to focus on Jewish mothers was made due to the significant differences observed between Jewish and Palestinian-Arab women in terms of their participation in the labor market and hence in their occupational pension. Fifteen of the mothers were divorced, two were widowed, one was single, and thirteen were married. Twenty-two of the women had a college education, while the rest had completed high school and had taken various courses over the years.

Table 3 provides the employment data for the group of interviewees. They had varied occupational histories and came from diverse places of residence and professions. Although most interviewees defined their pension as sufficient for daily living, a significant number (nine) continued to work (usually part-time) after they officially retired (twenty-nine percent). Ten of the women retired several years earlier before the statutory retirement age. There were also those whose economic status was particularly fragile: their pension does not provide for subsistence, nor they do not own an apartment. A few women had a relatively strong economic status; even when their pension benefits were not large enough, they found other sources of investment to improve their finances in old age. Eleven women took advantage of pension counseling and claimed that it helped them understand more about their future financial situation upon retirement.

The interviews were conducted in 2021–2022, during much of the COVID-19 pandemic. Individual interviews, lasting from one to two and a half hours, were mainly held in the participants’ homes, although some were held via Zoom during the COVID-19 waves. All interviews were conducted by the author, who introduced herself and explained the research project, guaranteeing the respondent’s confidentiality (see [125]).

All interviews were recorded, transcribed, and, following grounded theory, analyzed thematically [124]. Multiple readings were performed and, based on systematic repetition [126], units of prominent meaning were identified (main themes). Each interview was analyzed twice—once after it was conducted, and again once the themes were extracted (see [127]). Analysis was stopped after the first ten interviews, at which time new themes were examined; this process was repeated every ten interviews until saturation was reached (see [128]). 

## 4. Findings

The findings revealed five main themes—economic abuse: an unequal distribution of pension funds upon divorce; regrets over past choices; the COVID-19 pandemic and pensions; the state’s responsibility for old-age economic security; and knowledge is important, and I can help others.

### 4.1. Economic Abuse: An Unequal Distribution of Pension Funds upon Divorce

Israeli law stipulates that there must be an equal distribution between divorced spouses of all property and resources accumulated together, including money in each pension fund. Nonetheless, the theme of unequal distribution of pension funds between ex-partners emerged from the stories of divorced women when reporting their past actions and the present implications of such actions. 

Several women spoke painfully of the financial concessions they were forced to make in order to obtain the divorce, as well as the consequences of those concessions later on in their lives. For instance, Natasha stated that she “can’t grow old in peace” and explained why: 


*“We were married for about twenty years, and we have four children, but when he realized I wanted to move forward with the divorce, he activated all possible concealment tactics: said the property was mortgaged and we had bank debt, hid the information on his pension funds. Luckily I did a comprehensive study of all our assets and money before I set out on the divorce path. But I believed him about the pension funds and didn’t check on them.”*
(Natasha, works as a laboratory worker, divorced, 59 years of age, and has four children).

As she has explained, Natasha’s divorce cost her a deteriorating economic status that intersected with her vulnerability at retirement age. In between these lines unfolds a story of economic abuse: her ex-husband hid pension funds from her, and this caused her further deterioration in her economic status at retirement age. Her situation reflects economic abuse involving the man’s deception in hiding shared financial resources, which is particularly harmful in the case of divorce [20].

When another participant, Ora, was asked whether she shared with her ex-husband all their joint resources, she replied in the negative:


*“He wouldn’t give a divorce, he didn’t, he wouldn’t give me, he wouldn’t give me. Look, all the time he whined, “I don’t have, I don’t have, I don’t have.” I don’t believe him, I don’t believe him. Because I did all the expenses, all the expenses [of the household]. Although he paid a little but […] I, with my poor salary, managed to do so many things. And he received huge compensation [which is part of the pension fund] from his workplace. He worked at […], it’s a lucrative place.”*
(Ora, works for an NGO that runs clubs for the elderly, divorced, 67, and has four children).

Four times in her interview, Ora repeated that if she had stood up for her financial rights, her ex-partner would not have let her divorce him. It was clear to her that she would have to surrender certain financial resources that she deserved in order to exit the marital relationship. This decision required her to continue to be employed at age 67 in order to survive, as her limited employment pension benefits only covered her rent. Her story echoes the studies pointing to the male abuser who maintains control of the family finances, money, goods, and assets, and keeps the wife “unbanked” [18,19,91,94]. By reducing Ora’s ability to manage, obtain, use, and maintain her economic resources independently, her ex-husband threatened her financial security as well as her potential to support herself later in life [20,94].

In summary, even though the law formally protects them from losing resources after divorce, several of the interviewed women withdrew from this struggle, often to protect their children. Economic abuse by their partners is usually hidden from view; it is directed at the women who “chose” to give up resources. In practice, this is not their choice, but rather a matter of coercion. This is consistent with the findings that, although the Israeli law calls for equal gender division between the husband and wife during the divorce process, in practice, women leave with significantly fewer resources than men [100]. The women whose stories were presented above told an economic story of a lack of resources, and the significant price they pay in old age as a result. 

### 4.2. Regrets over Past Choices

Many of the women regretted their past conduct with respect to pension savings. They pointed to their inability to correct their situation in retrospect and felt they should have positioned themselves differently in their employment and couplehood negotiations. As a result of their choices when they were younger, they are now exposed to a vulnerable economic status, and cannot offer financial support to their children as a consequence. Such support is particularly needed under current economic conditions, which make it difficult to cope with the day-to-day expenses, especially with the rising cost of housing in Israel.

As mentioned earlier, Natasha chose not to fight her ex-husband over pension funds when she was younger. Now, close to retirement age, she recognizes the economic implications of that choice:


*“I was young, in my early forties, and I said to myself, “I pretty much do not have the strength for struggles; I’ll manage. I have many more years of work to fix it.” Today I realize that was a mistake. It can’t be corrected. And it hurts, that you can’t grow old in peace.”*


Moreover, as she did not negotiate for equal resources during the divorce, her children are now also paying the price:


*“Now, retroactively you can’t fix it, and it’s unfortunate that you can’t give your children financial help as you would like because when you were young, you sacrificed. I don’t have to tell you how costly it is to live in Israel and how expensive housing is.”*


Natasha’s story reveals how differently things look at different stages of the life course. When she was younger, she was unable to grasp the long-term consequences of her decision to avoid negotiating with her ex-husband over their shared resources. Since then, she now understands and painfully expresses the limitations of her earlier way of thinking as she looks at her present financial picture and that of her children, especially in light of the increased life expectancy, housing prices, and everyday costs in Israel.

Many of the women interviewed worked at a time when pension contributions were not mandatory (prior to 2008). Consequently, most did not save up for retirement and were left with the very low old-age pension granted by the National Insurance Institute. Alma (another interviewed participant) is an example. She has to still work for very low wages at the age of 72, as she has no employment pension, and subsists merely on her old-age pension. She described her economic struggle, as well as her sorrow in not being able to support her children:


*“It’s really not enough. 4000 shekels [~$1090; her salary] plus 2100 shekels [~$572; old-age pension] is 6100 shekels. It’s not enough to live on. Not to mention helping kids or something, absolutely not.*

*Q. I see that this point of helping your children is terribly painful for you?*

*A. Yes, definitely, because I grew up with middle-class parents. In those times the middle class could very much support their kids, and my parents supported me a lot. I mean, we bought an apartment with a mortgage but they gave us an amount. And every time we were in a less favorable financial situation during the marriage they always helped. I mean, I’ve very much aware of parental help. […] Obviously, I would be happy if I could help them [her kids] with that.”*
(Alma, caregiver working through an employment agency, divorced, 72 years of age, and has two children).

The women cited above express regret over the lack of longitudinal planning over their life course. When they were young, busy raising their young children, fighting for daily survival, and paying their rent or a mortgage, it was difficult to think and plan for their financial situation in old age. Now, later in life, they express their pain and regret for not taking sufficient care of both themselves and their children.

These findings resonate in respect to the unresolved debate in academic circles and international forums over whether saving for a retirement pension should be mandatory. Supporters of coercion argue that most young workers find it difficult to predict or plan for their financial future and therefore should be forced to save for retirement throughout their working years, in order to protect themselves from vulnerability to poverty and economic deprivation in old age [129].

### 4.3. COVID-19 and Pensions

The COVID-19 pandemic created financial difficulties for many of the interviewed women, which impacted their pensions. Havaselet, who is self-employed as a graphic artist (separated, 63 years of age, and has one child), relates how she has had to cope with a considerable drop in available work:


*“During the [height of] COVID, it was quite difficult for me.… For a large part of my work, customers had to make changes. There were several months without work. If I did [work] for one of my big clients—they were doing ceremonies…they had to cancel them and adapt them to online [ceremonies]. … Some clients even closed and stopped working…”*


Zohar speaks less about financial hardship and more of how the crisis made her consider early semi-retirement:


*“Women from age 62 can take their pension allowance, [or] they can postpone it for up to five years and earn five percent more for each year. Okay, I [chose to retire] because I’d just finished [teaching] twelfth graders and it suited me [to leave my position] at 64. I could have continued until 67.”*
(Zohar, works as a teacher, divorced and remarried, 64 years of age, and has three children).

Zohar indicates that she had already started thinking about retirement when she was 60 and the pandemic accelerated the process: 


*“The new principal for the last three years didn’t suit me…. I felt that if I continued it was already starting to cost me my health…. COVID made things even more extreme. I said [to myself] “come on, I’ve had enough!” Even so, towards the age of 60, I started all these reflections on how, what, when I’ll retired. So, I said I will finish [teaching] my twelfth-grade class and that’s it, that’s enough. …I’m still working, but elsewhere, only half-time, so that I have additional income to add to my pension allowance.”*


Liliana shared her story of losing her earnings during the COVID-19 pandemic, which at least entitled her to unemployment benefits, but found that they were lower than her usual earnings:


*“I’m unemployed. I really need to go to the labor office [to look for other work]. I haven’t gone until now because I. every time they canceled due to COVID. But I have to go now. I have to say that I’ve been receiving unemployment benefits, for the first time in my life, for a year and a half. … True, I didn’t get as much income as I normally do. But I have to say that it came in like clockwork every month for a year and a half…. I know many of my friends fell through the cracks and didn’t get anything, but I did. My daughter fell through the cracks because she was self-employed and finished her [last] job right before COVID, so she didn’t get anything.”*
(Liliana, currently an unemployed tour guide, married, 63 years of age, and has two children).

All the above stories point to the far-reaching economic consequences of the COVID-19 crisis, which affected, among other things, future pension benefits. Loss or complete cessation of work impacts the woman’s ability to contribute to their pension funds as well as their entitlement to the state’s old-age pension, which is determined by the number of work years.

In contrast, Maris relates how the COVID-19 crisis enabled her to improve her future pension benefits. Similar to the interviewees in the previous theme, she regretted her past choices with respect to retirement funds:


*“Now, my husband and I have a pension. It’s not big, even though I worked in a lot of places, but like some […] kibbutznik, excuse the expression, I never thought about a pension at all, I didn’t think about taking care of all these things. Our insurance agent, who started seeing it, said “Tell me, what is this? I need to arrange some 19 jobs for you, start building for you.” Never mind, a mess. But I didn’t think about it at all and I didn’t see it. And now it’s very important for me to have some kind of income if I want to buy anything for my grandchildren…”*
(Maris, a retired psychologist and part-time freelance cook, married, 68 years of age, and has three children).

Aware that she and her husband lacked adequate current income, Maris thought about “what I really want to do, what I like, what would be pleasant and good for me” and decided to become a part-time freelance cook:


*“I decided that I will post here, on the local Facebook, that I know how to cook well, and [ask] if families want me to come and cook for them. Two families responded. [In one family, they said] “The children are going to school, we want you to prepare food, on Wednesday they will return at noon from school, we want Is….” And it will be now, it’s all new. This is the first time. I didn’t sleep because I was stressed [hoping] that everything would be okay, and whether I would finish everything, whether it would taste good, and whether they’d be satisfied….”*


Thus, Maris offers a story of financial recovery thanks to the time that the COVID-19 pandemic gave her to rethink her income during retirement, and how she could find a solution to her insufficient income.

### 4.4. The State’s Responsibility for Old-Age Economic Security

As represented above, many of the interviewed women voiced their inability to plan for a retirement future when they were young and/or pointed to the increased financial difficulties posed by the COVID-19 pandemic. Most argued that the state should be responsible for allowing them to live with dignity in old age. This mainly revolved around the issue of contributions to an employment pension. Some countries have made it mandatory to contribute an amount towards such a pension, while others have not (OECD 2021a). As mentioned, this only became mandatory in Israel in 2008.

In her interview, Yaela argued that an employment pension should be mandatory, as evidenced by her personal case. As she did not understand the consequences of her financial decisions when she was younger, she is now paying a heavy price at retirement age:


*“The bottom line is that I have no [employment] pension. Because I took some of the benefits when I was fired. I had to resign from Dan [the bus company] because I could no longer continue working due to physical restrictions. I went on unemployment, but I also took worker’s compensation [funds which are part of the pension allowance base] and other benefits. But I didn’t save for retirement. I spent my pension. Not that there was much, but regardless, I spent it. I’ve worked for forty years and contributed added value to what I do. So why don’t I Ierve it…why does the so-called state hold me back? I say again that the bottom line of what I said before is that the state should ensure my pension; it shouldn’t fall on me to make sure it happens.”*
(Yaela, currently unemployed, divorced, 63 years of age, and mother of one).

Yaela believes that the financial security of the elderly is the state’s responsibility. In her view, the average person is focused on making a living and obtaining financial resources throughout their life with the aim of generating sustenance for themselves and their family. Pension savings, on the other hand, are much more complex. People need to be guided through this issue. The state should not make them solely responsible for their own financial security in old age.

Sima also emphasizes her inability to anticipate her retirement future when she was a young worker and calls upon the state to take responsibility for the elderly’s economic security:


*“I have no pension [laughs]. It’s not so simple then. I mean, my pension…what I managed in places where I was a salaried employee [most of the years she was self-employed] here and there will give me a employment pension of 150 shekels [~$41] a month. So I have no employment pension […]. That’s wrong. So it’s not that I felt I had to chose a different line of work, but rather that the state is wrong: That’s not how they should operate; it’s not right. Loads of people are thrown out [of the system]. I don’t think there’s anyone in the film industry [where she worked in the past] who is set up for retirement. Maybe someone who’s reached the top and has too much money. The film industry in Israel does not earn that much money. So that’s it. And even now, as I talk to you, I don’t feel like I should’ve behaved differently and arranged a pension for myself. I’m not succeeding [now] because I fought to have enough food that month. So pension: who thought of that at all?”*
(Sima, works as a freelancer in knowledge management, widow, 63 years of age, and has one child).

Later in the interview, Sima concludes: “In my eyes, the [employment] pension should be part of the National Insurance system and a person should really be taken care of as soon as he cannot work.” In other words, the state should be responsible not only for old-age allowances, but also for employment pensions. 

In a similar vein, Maya points to the importance of state responsibility for the self-employed:


*“Most self-employed have never learned to manage their budget, manage their cash flow. So the same, they also don’t manage their future pension. So the matter of the patronage of the state is necessary because these people are not thinking about their future.”*
(Maya, works as a self-employed business consultant, married, 60 years of age, and has three children).

In reference to how difficult it is for low-wage workers to save for a pension and the burden this creates in terms of their daily survival, Maya states: 


*“I think that, really in these cases where the salary is so low, the state should contribute the [employee’s] part [of pension savings]. The employer must contribute his part, it’s mandatory, […] but the employee’s part the state should see to.”*


In Maya’s eyes, the matter cannot be left solely to the individual. The state needs to take responsibility for savings. She even proposed a more radical solution for low-paid workers: the state should pay their contributions to a pension fund. In other words, Maya requests that the state not abandon people to their own fate, and to be an active player in ensuring economic security in old age.

In summary, the interviewed women point to the inability of young workers to predict and plan their future at retirement age. They call upon the state to take responsibility for elderly economic security: to plan the pensions of its citizens throughout their years of employment from an early age, and to ensure a reasonable economic status for them in retirement and old age. In their eyes, the state should be caring and responsible for its citizens. Instead, it is currently penalizing low-income and poor old people who lacked the wherewithal to properly plan their financial future when they were younger. In their opinion, this is too great a burden to place on the individual; rather, it should be a public responsibility.

### 4.5. Knowledge Is Important and I Can Help Others

Many of the women pointed to the importance of their knowledge regarding pension funds, especially of the relatively new (and complex) financial tools. Some went a step further, taking responsibility for the economic wellbeing of others, explaining, and supporting ways to ensure more financial security at retirement age. 

Recounting her father’s advice on the importance of pension funds, Maya relates how it has shaped her thinking about a secure financial future:


*“So [my father] used to say: “Daughter [laughs], it’s very important that you always have a pension fund. Savings and finances do not matter. This pension fund is important. You need to have an amount that goes into your account every month.” And, of course, I also studied the issue in my work and also worked with agents, etc. I realized he was right. I also started talking to whoever I could and with employees in particular, explaining the importance of this matter. That it is very important to have not only a capital amount but also an annuity that we will receive in the future. When it was not yet mandatory, it was before the so-called mandatory pension.”*


Furthermore, she tells of her drive to help anyone who will listen to obtain this knowledge:


*“I tried to instill in people that it’s important to have some minimum amount that will go into their bank account every month, that it will not depend on anything, and that it can give them some kind of security even in old age.”*


Anna explains why she believes a pension training course should be given to everyone at a young age: 


*“They need to be given a course like this at university, like a mandatory course, […] or in the workplace, workplaces should provide such a course. It’s for the employee’s wellbeing, really. This is an ongoing course. When I was with my husband on a retirement course, I said to him, “Wow, why didn’t they give this to me ten years ago, twenty years ago, why now?” I mean, what am I going to do with [this information] now? Okay, let’s take a risk, but how much will it help me right now? I don’t have thirty years now [to accumulate savings]…. There are loads and loads of financial tools today that people are not aware of, really. And acquiring them as soon as they retire is too late […] So we need to give these skills to women […]. So one has to give these skills in school as well.”*
(Anna, currently self-employed in international marketing, married, 64 years of age, and has three children).

Anna speaks passionately about the importance of the knowledge needed to cope well with one’s economic future at retirement age, giving her own situation as an example. She points to the state’s responsibility for continuously imparting the knowledge needed to ensure people’s economic security in old age. 

Galia also emphasizes the importance of imparting knowledge:


*“I think there are a great many women, probably my age […] but not only—but also younger people—who don’t exactly understand what it means to have a pension and its implications. Whenever people come here [to her accounting office] and I sometimes hear what kind of economic steps they want to take, I always stop them and say: “Wait a second, stop. Does this make sense?” It’s not that you have money here and now you’re on top of the world. It has to do with doing something with that money. It needs to support you. I always shamelessly give the example of my mother and my father. Just a month ago, my mother sold her apartment. It’s shocking to me that an elderly person has to sell her home in order to survive…and I’m amazed over and over how it still is… young people who are drowning in material possessions and employees who still don’t understand what it means to need money at an older age. […] I’m here doing what I can, but yes, I’d like there to be more explanation of what this thing called pension means. Not everyone understands it.”*
(Galia, works as a payroll accountant, divorced, 60 years of age, and has four children).

Finally, Miri relates how she has learned, from her work as a human resource manager, how different pension compositions have different effects on employees’ financial futures. She explains how she decided to go beyond her formal role and invest her time and effort in each employee to protect everyone’s pension funds:


*“While employees were signing employment agreements, I also checked their pension amounts and discovered that everyone, from senior vice president to cleaner, has the same pension plan insurance, where 28 percent goes to risk [insurance in case of death] and 72 percent goes to the pension. Intuitively, I said to myself this can’t be, it doesn’t make sense that different people with different marital statuses, different ages, different occupations and different incomes would have the same insurance. A lot of money is going into risk insurance. […] And since then, we really sat down, worker by worker, and I learned a lot and adapted plans for each worker. […] We need to understand, we need to check the matter of retirement savings every so often or whenever there’s a change in family or income status.”*
(Miri, retired, married, 67 years of age, and has two children).

The women cited above emphasized the importance of understanding the implications of pension savings in order to live with dignity in old age. This is in keeping with the theories on financial literacy, which argue that knowledge can have a positive effect on the individual’s financial quality of life [51]. However, experience shows that knowledge is not sufficient; structural mechanisms are needed to ensure people’s economic security. The women also pointed to the difficulties of navigating the everchanging pension options. This was deemed to be probably related to the move to complex business tools in the neoliberal era [20,49]. Finally, the women took the knowledge they had regarding financial security in retirement and passed it on to others. In so doing, they revealed their self-confidence in their abilities and their desire to help others.

## 5. Discussion

Based on the in-depth interviews of thirty-one mothers aged 59–72, and adopting a life course perspective, this study examined the women’s retrospective view of their pension planning when they were younger, its implications for them in the present, and their perceptions of the state’s role, against the background of the COVID-19 pandemic. The focus was on the accumulation of major life events and experiences (e.g., divorce and economic abuse) at their intersections in women’s lives [1,3,4,5]. As such, the current research addressed gaps in the literature on the intersection between the women’s employment history, their vulnerable economic retirement status, and their marital and parental status. These vulnerabilities have been aggravated by the COVID-19 pandemic, as lockdowns and closures deepened the gender inequality [80,81,82,83] and employment dropped substantially more than for older workers [84].

Findings about the retrospective and present views of older mothers regarding their pension planning pointed in two directions. First, divorced women expressed remorse regarding their inability to negotiate with their ex-husbands over pension funds and other economic resources when they were younger, which was found to be indicative of economic abuse at the time of separation. Such economic abuse prevents women from accumulating their resources [20], threatens their financial security, and hinders their potential to support themselves [18,19,91,92,93,94]. Therefore, this may have critically affected their ability to survive in old age. Indeed, the women reported the negative consequences of their past choices, which aggravated their fragile economic status in the present and future and prevented them from financially assisting their adult children. These stories point to another arena of women’s vulnerability that leaves them depleted of resources as they reach retirement age.

Second, the study findings emphasize the economic vulnerability of divorced mothers in old age. This supports the argument that cumulative vulnerability over the life course may shape financial fragility in old age [4]. Women tend to become economically vulnerable following divorce [11,12,13,14,15,16], and this vulnerability is magnified during retirement [8,9,10,40,41,42].

With respect to the mothers’ perceptions of state pension policy, again, two main directions emerged from the findings. First, in the women’s understanding, it is the state’s responsibility to guarantee economic security for its citizens in their old age. They believe that a pension is not a private issue, but rather a public matter. In their opinion, the state abandons individuals to their own fate and shirks its responsibility. They attribute the financial vulnerability of retired women [8,9,10,24,25,26,27,28,29,30] to a lack of state responsibility for the pension system and the economic rights of the elderly. Indeed, studies have shown that the Israeli pension policy leaves the risks and responsibility for economic survival at retirement age to the individual whilst widening the gender and socioeconomic gaps [23]. This was further exacerbated by the loss of jobs during the COVID-19 pandemic, which led to a rise in retirement among the older individuals and women [86]. Early retirement can reduce the pension benefits and raise income inequality in later years [87,88].

Second, considering the state’s inaction, the mothers have come to the conclusion that there is no one to provide the knowledge about financial needs at retirement age, and thus they are stepping forward and sharing with others the knowledge they have accumulated over their life course and work history. This knowledge translates into power, creating a sort of “maternal citizenship for retirement,” whereby they help other women avoid the mistakes that cannot be corrected when they reach old age.

The life course perspective that this paper adopts (following [1,2,3]) has highlighted the intersections of women’s vulnerabilities over time, and particularly of how they are experienced in post-retirement. Life course outcomes are often subjected to accumulated disadvantages, as early life difficulties can be compounded as time passes [5].

Listening to the women’s pension stories, we can learn from them on how events and experiences that have occurred throughout the life course shape financial fragility at retirement age, creating a vulnerability that can no longer be repaired. The mothers argue that pension savings are a public matter for which the state must take responsibility; otherwise, gender and class differences are reproduced and magnified as people age. Their call for state responsibility for pensions echoes Ben-Israel’s [129] claim that, in light of the short-sightedness of young people about their financial future, it is up to the state to prevent the financial deterioration of the elderly, especially of older women.

### Research Limitations and Practical Recommendations

The study is not without limitations. The group of mothers who were interviewed is limited in size, although it was deemed to be appropriate for qualitative research, and it overrepresents divorced and educated women. Nonetheless, it does provide a deep knowledge of the experiences of the interviewees, which complements the existing quantitative research of larger samples [10,28,46] arguing that women in retirement are much more vulnerable than men. Future studies of a larger, more representative population would provide an even more diverse group in which to examine the financial situation of women approaching retirement. Further research should examine other groups in Israeli society, such as Palestinian-Arab, immigrants, and ultra-Orthodox women.

Notwithstanding this limitation, the study reveals the complex circumstances that mothers have to face at retirement age. The findings suggest that it would be valuable to deepen the state’s responsibility for women’s retirement. For example, the state could implement a policy that compensates them for unpaid care work to ensure that caregiving does not harm their economic future. Additional protections could also be given to divorced women in terms of the enforcement of pension distribution between spouses.

## 6. Conclusions

The present study makes several contributions to the literature on women and retirement. Its focus on the intersections of gender, vulnerability in old age, motherhood, and the life course in the context of employment pensions, makes it innovative at the theoretical level. Furthermore, by taking a unique qualitative approach, the empirical findings obtained from this study enrich the results of quantitative feminist studies [10,28,46], which have claimed that the position of women in retirement is much more vulnerable than for men, thereby requiring intervention at both local and international levels. Specifically, the study’s in-depth qualitative methodology, which focuses on the voice and experiences of the women, has revealed the perceptions of mothers about their economic status at retirement age. The findings highlight how, in light of the women’s more convoluted and vulnerable life path, that it is up to the state to take an extended responsibility and provide women with added protections. This requires creative solutions, such as what one interviewee suggested—namely, that the state finances poorer workers during their working years with an employment pension. Finally, the interviews suggest that pensions are not a private matter, but rather a public one. Similar to other spheres in which feminists claim the personal is also public, here, too, the poverty and economic deterioration of women in retirement appears to be a matter for public policy.

## Figures and Tables

**Table 1 ijerph-20-05912-t001:** Sources of retirement income in Israel: a three-tiered structure.

Old Age Allowance (National Insurance)	Employment Pension Based on Earnings	Private Pension Based on Personal Savings
Determined by the National Insurance Law. Universal; eligibility based on permanent residency. Women of age 62–65, (retirement age for women has been rising gradually, depending on the year of birth), and men of age 67. Means tested up to age 70. All residents above age 18 must set aside a minimum monthly insurance payment even if they are not employed.	Mandatory for all wage workers, employees, and the self-employed. Based on a legal arrangement (not a law) from 2008. Some provisions qualify for a tax credit.	Non-coercive, based on the voluntary provisions of employees, employers, and the self-employed. Such arrangements as provident and training funds can be used as a pension allowance. Some provisions qualify for a tax credit.

**Table 2 ijerph-20-05912-t002:** Monthly universal old-age pension from the National Insurance (2022).

	Up to Age 80		From Age 80
Base rate (there is a seniority increment of 2% for every year the National Insurance tax was paid, up to 50%; and a pension deferral increment of 5% for every year after retirement age that receipt of the allowance was deferred, up to 25%.)	1596 NIS ($435) (The exchange rate is calculated according to $1 = 3.67 NIS.)		1686 NIS ($459)
	Up to age 70	Ages 70–79	From age 80
With income support supplement	3799 NIS ($1035)	3836 NIS ($1045)	3873 NIS ($1055)

**Table 3 ijerph-20-05912-t003:** Employment characteristics of the participants (*N* = 31).

Variables	*N*	%
Employment status		
Employed	22	71.0
Retired (or semi-retired)	9	29.0
Occupation (now and before retirement)		
Administrator	6	19.3
Education	4	12.9
Human resource manager	3	9.6
Therapist	5	16.1
Architecture and design	2	6.4
Finance management	2	6.4
Other (tour guide, director, library scientist, and dentist)	9	29.0

## Data Availability

Data is unavailable due to privacy or ethical restrictions.

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
