# Peer review of "What’s Your Pension Story? Women’s Perspectives during the COVID-19 Pandemic on Their Old-Age Pension Status, Past and Present"

_ijerph, 2023, doi:10.3390/ijerph20105912_

Round 1

Reviewer 1 Report

Manuscript ID: ijerph-2231077

The manuscript deal with an interesting and important issue.

It is important to understand the difficulties and limitations facing adults in general, and women in particular, in regard to their perception of the pension situation, pension policy and proper pension planning. Therefore, I find great importance in the study. However, the manuscript suffers from some critical problems.

It is certainly possible to understand why the subjective perception regarding economic status and pension plans are related to various events during life such as the employment history and the course of the career, family status, economic status, health status, etc. But it is not clear to me how and why the COVID 19 is related to women's perception of their pension plans. The connection between these two variables seems forced and unclear. Economic abuse and vulnerable situation of women in the labor market is not necessarily related to the COVID 19 period. Their vulnerable situation in the labor market and in the family, which depends on and is influenced by many factors and variables, existed even before the pandemic.

Literature Review:

In the literature review there is one small paragraph that deals with the COVID 19.  It looks disconnected and out of place. Therefore, I propose one of the two options: canceling the paragraph, and accordingly make the necessary changes in the entire manuscript. Another possibility is to explain in a more detailed and convincing way why the COVID 19 pandemic affects women's perception of their pension plans. For example, did the COVID 19 affect the decision to retire early? Paragraph 2.3 cannot constitute a chapter.

Chapter 2.4: how is the economic abuse against women expressed in the family and the labor market? Are there differences in economic abuse between married and divorced women.

What is the situation of elderly widows? Do they also suffer from economic abuse by family members? Are they aware of their pension rights? Are there differences between widows and divorcees? (Although there is only one widow in the sample, it was interesting in the literature review to refer to this issue as well).

Are there differences between the women according to the age at which they retired? That is, did women who decided on early retirement, before the retirement age defined by law, had pension planning? Did they understand the pension policy better or less than those who retired when they reached retirement age or after the retirement age defined by law?

Sample description:

Additional details are missing for the sample description. Are all the interviewees Jewish? What is their economic status? education level? What was their profession and occupation before retirement? How many of them retired early, before the retirement age defined by law?

Fourth theme:

Did the women receive pension counseling and detailed explanations regarding their financial situation and what are their pension rights? How many of them were helped by an expert or a pension consultant in the field? Was the referral to pension counseling their initiative? How much did it help them in their pension planning?

Discussion:

The discussion is too short. Since the life course theory is mentioned in the literature review, it is appropriate to return to it in the discussion and examine how it contributes to our understanding of the way in which women perceive their economic status and pension planning.

Research limitations and practical recommendations: What are the shortcomings of the research and what can be done in practice to improve and change the vulnerable situation of women, regarding to their pension planning and pension policy?

There is no closing paragraph. There is a feeling that the article is cut in the middle and there is no closure.

Author Response

Dear Reviewer,

Thank you very much for encouraging me to revise and resubmit my manuscript,
Manuscript ID: ijerph-2231077, entitled “What’s Your Pension Story? Women’s Perspectives During the COVID-19 Pandemic on Their Old-Age Pension Status, Past and Present.” I have incorporated most of the reviewers’ comments and believe the manuscript is significantly improved as a result.

Revisions are marked in red in the text. Please find below (also in red) a detailed explanation of how I addressed each comment. I will be happy to provide any further clarification as needed. Finally, I would like to take this opportunity to thank you for your helpful feedback.

Yours sincerely,

The manuscript deal with an interesting and important issue.

It is important to understand the difficulties and limitations facing adults in general, and women in particular, in regard to their perception of the pension situation, pension policy and proper pension planning. Therefore, I find great importance in the study. However, the manuscript suffers from some critical problems.

It is certainly possible to understand why the subjective perception regarding economic status and pension plans are related to various events during life such as the employment history and the course of the career, family status, economic status, health status, etc. But it is not clear to me how and why the COVID 19 is related to women's perception of their pension plans. The connection between these two variables seems forced and unclear. Economic abuse and vulnerable situation of women in the labor market is not necessarily related to the COVID 19 period. Their vulnerable situation in the labor market and in the family, which depends on and is influenced by many factors and variables, existed even before the pandemic.

Thank you very much for your helpful feedback. My point is that the pandemic accentuated these vulnerabilities and often caused the difficulties to surface earlier. See also my response below.

Literature Review:

In the literature review there is one small paragraph that deals with the COVID 19.  It looks disconnected and out of place. Therefore, I propose one of the two options: canceling the paragraph, and accordingly make the necessary changes in the entire manuscript. Another possibility is to explain in a more detailed and convincing way why the COVID 19 pandemic affects women's perception of their pension plans. For example, did the COVID 19 affect the decision to retire early? Paragraph 2.3 cannot constitute a chapter.

Thank you very much for this important comment. I chose the second option you suggested and expanded the section on the consequences of COVID-19 on gender and socioeconomic inequality, as well as on retirement (pp. 6-7).

Chapter 2.4: how is the economic abuse against women expressed in the family and the labor market? Are there differences in economic abuse between married and divorced women

I have added a few sentences about the connection between economic abuse and divorce (p. 8).

What is the situation of elderly widows? Do they also suffer from economic abuse by family members? Are they aware of their pension rights? Are there differences between widows and divorcees? (Although there is only one widow in the sample, it was interesting in the literature review to refer to this issue as well).

I have added a paragraph on widows (p. 6).

Are there differences between the women according to the age at which they retired? That is, did women who decided on early retirement, before the retirement age defined by law, had pension planning? Did they understand the pension policy better or less than those who retired when they reached retirement age or after the retirement age defined by law?

While the questions you ask concerning knowledge about pensions are interesting and of value, they are beyond the scope of the present paper. The study does not focus on financial literacy, but rather on structural constraints upon women’s retirement and financial security in old age.

Sample description:

Additional details are missing for the sample description. Are all the interviewees Jewish? What is their economic status? education level? What was their profession and occupation before retirement? How many of them retired early, before the retirement age defined by law?

These details have been added (pp. 11-12 and Table 4).

Fourth theme:

Did the women receive pension counseling and detailed explanations regarding their financial situation and what are their pension rights? How many of them were helped by an expert or a pension consultant in the field? Was the referral to pension counseling their initiative? How much did it help them in their pension planning?

There is additional information about this in the Method section (p. 11-12).

Discussion:

The discussion is too short. Since the life course theory is mentioned in the literature review, it is appropriate to return to it in the discussion and examine how it contributes to our understanding of the way in which women perceive their economic status and pension planning.

I have expanded the discussion and added a paragraph on life course to the Discussion (pp. 25-26, 27-28).

Research limitations and practical recommendations: What are the shortcomings of the research and what can be done in practice to improve and change the vulnerable situation of women, regarding to their pension planning and pension policy?

I now refer to study limitations and raise practical recommendations emerging from the study findings, with suggestions for reducing the harm to women (p. 28).

There is no closing paragraph. There is a feeling that the article is cut in the middle and there is no closure.

Thanks for this comment. I have added a Conclusion section (pp. 28-29).

Reviewer 2 Report

Dear Author,

The article deals with the important issue of women's pension security. The topic addressed is therefore important and timely. It has the added advantage of looking at the issue in the perspective of the Covid 19 pandemic. However, the scientific value of the article is small. The author conducted a study on a group of 31 mothers. This is an insufficient sample to conduct reliable scientific research, which discredits the scientific value of the results. 

Moreover, the author did not explain how the women were selected for the survey to ensure the representativeness of the sample, and did not conduct any analysis that would have scientific value. The article consists solely of quoted statements from the women surveyed. The conclusions drawn are therefore not supported by factual arguments. 

An additional note: the cited literature is overly extensive. Such a large number of items adds nothing substantive to the content of this prticular article. The author does not argue with or critically address any previous statements.

Author Response

Dear Reviewer,

Thank you very much for encouraging me to revise and resubmit my manuscript,
Manuscript ID: ijerph-2231077, entitled “What’s Your Pension Story? Women’s Perspectives During the COVID-19 Pandemic on Their Old-Age Pension Status, Past and Present.” I have incorporated all the reviewers’ comments and believe the manuscript is significantly improved as a result.

Revisions are marked in red in the text. Please find below (also in red) a detailed explanation of how I addressed each comment. I will be happy to provide any further clarification as needed. Finally, I would like to take this opportunity to thank you for your helpful feedback.

Yours sincerely,

The article deals with the important issue of women's pension security. The topic addressed is therefore important and timely. It has the added advantage of looking at the issue in the perspective of the Covid 19 pandemic. However, the scientific value of the article is small. The author conducted a study on a group of 31 mothers. This is an insufficient sample to conduct reliable scientific research, which discredits the scientific value of the results.

As this study is qualitative,  a group of this size is sufficient to glean in-depth information on women’s experiences, perceptions and thoughts. In the manuscript, I now point to the sample size as a limitation, while also highlighting the advantages of such a research approach. 

Moreover, the author did not explain how the women were selected for the survey to ensure the representativeness of the sample, and did not conduct any analysis that would have scientific value. The article consists solely of quoted statements from the women surveyed. The conclusions drawn are therefore not supported by factual arguments. 

Thanks for the important comment. I have written more information about the sample and how it was recruited in the Method section (pp. 11-12). Moreover, I have tied the discussion more closely to the studies and theories cited in the Literature Review.

An additional note: the cited literature is overly extensive. Such a large number of items adds nothing substantive to the content of this prticular article. The author does not argue with or critically address any previous statements.

I have revamped considerable parts of the literature review, removing references that were not very relevant and adding new ones.

Round 2

Reviewer 1 Report

The manuscript was significantly edited in light of the comments. The main idea is presented more clearly, and the topic is interesting both in the Israeli context and general. The article should be published.

Author Response

April 20, 2023

Dear Reviewer,

Thank you very much for encouraging me to revise and resubmit my manuscript,
Manuscript ID: ijerph-2231077, entitled “What’s Your Pension Story? Women’s Perspectives During the COVID-19 Pandemic on Their Old-Age Pension Status, Past and Present.” Again, thank you all for your helpful feedback.

Yours sincerely,

The manuscript was significantly edited in light of the comments. The main idea is presented more clearly, and the topic is interesting both in the Israeli context and general. The article should be published.

Thank you very much for the supportive review.

Reviewer 2 Report

Dear Authors,

You have put a lot of effort in correcting the theoretical part. The literature part has been significantly improved, but my opinion on the research part remains unchanged. The research sample that entitles to draw reliable conclusions is insufficient. Moreover, the selection process is very perfunctorily described. There is a high probability of biasing the research results. Conclusions based on the opinions of a few people have no scientific value. I rate the scientific value of the work low .

Author Response

April 20, 2023

Dear Reviewer,

Thank you very much for encouraging me to revise and resubmit my manuscript,
Manuscript ID: ijerph-2231077, entitled “What’s Your Pension Story? Women’s Perspectives During the COVID-19 Pandemic on Their Old-Age Pension Status, Past and Present.” I have addressed your comments as detailed below. Revisions are marked in red in the text. Again, thank you all for your helpful feedback.

Yours sincerely,

You have put a lot of effort in correcting the theoretical part. The literature part has been significantly improved, but my opinion on the research part remains unchanged. The research sample that entitles to draw reliable conclusions is insufficient. Moreover, the selection process is very perfunctorily described. There is a high probability of biasing the research results. Conclusions based on the opinions of a few people have no scientific value. I rate the scientific value of the work low.

In the current revised version, I have revamped and reorganized the Method section to address your concerns (pp. 11-12). Thus, I emphasize the importance of qualitative research. Such research is not measured by the same criteria of validation and quantity as quantitative research, but carries other advantages. Following Small (2009, p. 28), my approach calls for “logical rather than statistical inference, for case rather than sample based logic, for saturation rather than representation.” Thus, though the sample was neither representative nor statistically large enough from which to generalize (Karazi-Presler, Sasson-Levy and Lomsky-Feder 2018), statistical representativeness was not relevant for my purposes (see Small 2009; Charmaz 2014).

In keeping with your request, I now also provide more details on the selection process and indicate that I reached a point of saturation. In addition, I have expanded the Limitations section to address the representativeness of the sample, as well as the contribution of my in-depth study findings to existing quantitative studies with larger samples. I also emphasize that a larger future study would enhance the revealed knowledge. Finally, in light of your comment, I have toned down my practical policy recommendations and conclusions (pp. 28-20).
